# Quantum refrigeration powered by noise in a superconducting circuit

Simon Sundelin ⊠, Mohammed Ali Aamir, Vyom Manish Kulkarni, Claudia Castillo-Moreno & Simone Gasparinetti ⊠

While dephasing noise often hinders quantum devices, it can become an asset for quantum thermal machines. Here we demonstrate a three-level thermal machine that leverages noise-assisted quantum transport to enable steady-state cooling of microwave modes. The device exploits symmetry-selective couplings between a superconducting artificial molecule and two physical heat baths. Each bath consists of a microwave waveguide populated with synthesized quasithermal radiation. Energy transport is enabled by injecting dephasing noise through a third channel longitudinally coupled to one artificial atom of the molecule. By varying the effective temperatures of the reservoirs and measuring photonic heat currents with sub-attowatt resolution, we demonstrate energy flow dynamics characteristic of a quantum heat engine, thermal accelerator, and refrigerator. Our work constitutes an experimental demonstration of the key operating principles of a noise-assisted three-level quantum refrigerator and opens new avenues for experiments in quantum thermodynamics using superconducting circuits coupled to physical heat baths.

As quantum technologies advance, so does the imperative to understand energy flows at the quantum level. The extension of thermodynamics to single quantum systems has uncovered fundamental insights into nanoscale out-of-equilibrium systems and the second law of thermodynamics[1-10]. This progress has also unveiled new avenues for boosting the efficiency of batteries[11] and for the optimization of quantum heat engines[12-16]. Of particular interest is the category of thermal machines that operate autonomously by harnessing heat flows to perform useful tasks[17]. In the quantum domain, these machines provide an ideal setting for measuring the thermodynamic cost associated with operations such as timekeeping[18,19], entanglement generation[20,21], and refrigeration[22-36].

In many quantum technologies and experiments, cooling serves as an essential preliminary step. Consequently, quantum absorption refrigerators represent a notable subset of thermal machines, distinguished by their capacity to provide cooling solely through the utilization of heat as a resource. The primary goal of such a refrigerator is to transfer heat from a cold bath of temperature $T_c$ to a hot bath of

temperature $T_h > T_c$ by using heat from a work reservoir at a temperature $T_w > T_h$[33]. The earliest and arguably simplest envisaged model of a quantum absorption refrigerator was based on a three-level system[37-39]. Recently, a more intricate three-body refrigerator was realized using trapped ions[30] and superconducting qubits[40].

Within the class of quantum refrigerators there exists another type of the autonomous variety, the Brownian refrigerator[41,42]. Analogous to the operation of Brownian motors[43-45], these devices facilitate unidirectional heat transfer in response to random noise. However, the development of a cooling system based on this principle has proven to be elusive because of the difficulty in harnessing noise as a resource. In the quantum domain, this form of energy transport is akin to noise-assisted excitation transport observed in quantum networks[46-49]. Such transport is believed to be a governing mechanism in the process of photosynthesis and its viability has been investigated in a three-site network[50]. Beyond leveraging this mechanism to function against a temperature gradient, demonstrating a noise-assisted refrigerator necessitates precise

Department of Microtechnology and Nanoscience, Chalmers University of Technology, Gothenburg, Sweden. ⊠e-mail: simsunde@chalmers.se; simoneg@chalmers.se

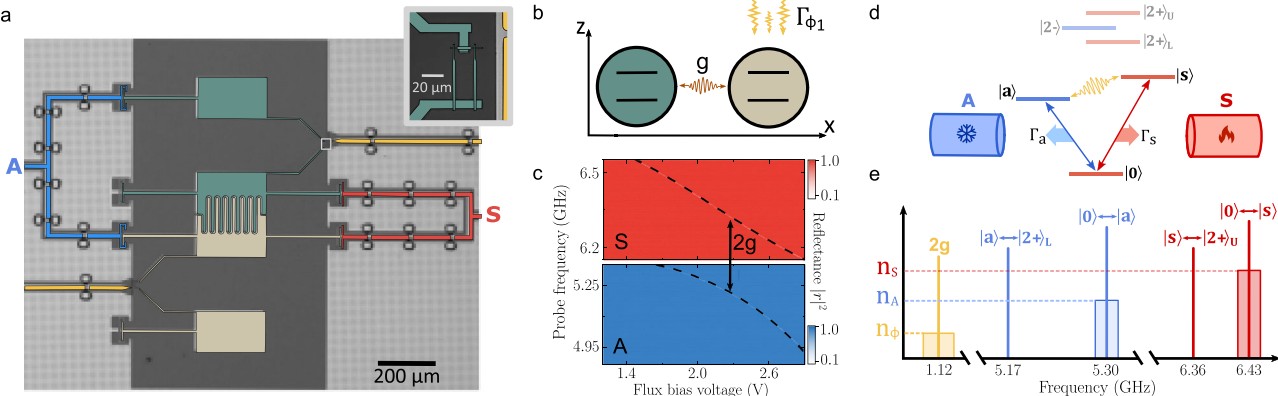

**Fig. 1 | Device architecture and energy level diagram. a** False-color micrograph of the device comprised of two frequency-tunable transmon, colored in green and beige, coupled to microwave waveguides labeled S (red) and A (blue). Flux lines coupling longitudinally to the system are colored in yellow. Inset shows a superconducting quantum interference device (SQUID) of one of the transmons. **b** The transmons represent two coupled qubits where each can be dephased through a longitudinally coupled channel. **c** Reflectance, $|r|^2$, through symmetric (S) and antisymmetric (A) waveguide as a function of applied flux voltage bias. The coupling rate, $g$, is obtained at the point where $\Delta = \omega_s - \omega_a = 2g$ is minimized. **d** Energy-level diagram showing states of even (red) and odd (blue) symmetry up to second-excitation manifold of the artificial molecule realized with the transmons. Symmetry-preserving (red) and symmetry-inverting (blue) transitions predominantly couple to waveguide S and A, respectively. For clarity, transitions to higher excited states have been omitted. The effect of the bare qubit dephasing in the collective state picture is to mix states $|a\rangle$ and $|s\rangle$. **e** Experimentally observed transition frequencies of the molecule in the first-excitation manifold along with the next closest transition. The red, blue, and yellow shaded boxes represents the spectral density of the injected heat baths corresponding to populations of $n_S$, $n_A$, and $n_\Phi$ in waveguide S, waveguide A, and one of the flux lines, respectively.

measurements of minute heat currents over varying temperature gradients, a technically challenging task.

In this work, we present an experimental realization of a superconducting device that exhibits the steady-state energy flows of an autonomous, noise-assisted three-level quantum refrigerator. Our system comprises an artificial diatomic molecule coupled symmetry-selectively to two semi-infinite microwave waveguides, which act as bosonic reservoirs. We set the effective temperatures of the relevant reservoir modes by injecting spectrally localized, calibrated noise around targeted transition frequencies[40]. Energy transfer between the reservoirs proceeds via noise-assisted excitation transport through the molecule, realized by injecting controlled dephasing noise on one of the artificial atoms. We directly resolve heat currents of microwave photons at the attowatt scale using simultaneous power-spectral-density measurements and interleaved power readout on each reservoir port. By tuning the dephasing and the relative reservoir temperatures, we observe energy-flow dynamics characteristic of a quantum refrigerator, heat engine, and thermal accelerator. In each case, the device induces a measurable change in the photon occupation of the microwave modes in the output fields it scatters, corresponding to their net cooling or heating.

## Results
### Symmetry-selective coupling
Our device is comprised of two nominally identical, flux-tunable transmon qubits[51], forming an artificial molecule. Each transmon consists of two superconducting islands which form the capacitor and are shunted by a superconducting quantum interference device (SQUID), to which we couple a flux line (Fig. 1a, b). The Hamiltonian governing the system can be expressed as

$$\mathcal{H} = \sum_{i=1,2} \omega_i(\Phi_i)\sigma_i^+\sigma_i^- + g\left(\sigma_1^+\sigma_2^- + \sigma_2^+\sigma_1^-\right). \quad (1)$$

Here $\omega_i(\Phi_i)$ are the bare, flux-tunable mode frequencies, $\sigma_i^+$ and $\sigma_i^-$ are the creation and annihilation operators of qubit $i = 1, 2$ respectively, and $g$ is the coupling rate between them[51]. When the bare mode frequencies are equal, the states $|10\rangle$ and $|01\rangle$ are resonant and in the molecule's single-excitation manifold they form the collective states

$|s\rangle = (|10\rangle + |01\rangle)/\sqrt{2}$ and $|a\rangle = (|10\rangle - |01\rangle)/\sqrt{2}$. These are symmetric and antisymmetric, respectively, and their frequencies are split by $2g$ [Fig. 1(d)]. Two microwave waveguides, denoted by S and A, are capacitively coupled to multiple points of the circuit to predominantly facilitate symmetry-preserving (waveguide S) and symmetry-inverting transitions of the molecule (waveguide A), with respect to a permutation of the two artificial atoms[52]. Hence, in the first excitation-manifold, the $|0\rangle \leftrightarrow |a\rangle$ transition largely couples to waveguide A, while the $|0\rangle \leftrightarrow |s\rangle$ transition couples to waveguide S. Simultaneously, flux lines longitudinally couple to the bare qubits, thereby creating the possibility to induce controlled dephasing through the application of noise.

To increase the sensitivity of the molecular modes to applied flux noise, we tune the transition frequency of qubit 1 away from its zero flux point ("sweet spot"). Next, we tune the magnetic flux in the second flux line until $\Delta = \omega_s - \omega_a$ is minimized (Fig. 1c). This point corresponds to the full hybridization of the qubits where $\omega_1 = \omega_2$, ensuring maximal isolation between the symmetric (antisymmetric) mode with the unintended antisymmetric (symmetric) waveguide. The frequencies of the resulting modes are determined to be $\omega_s/2\pi = 6.426$ GHz and $\omega_a/2\pi = 5.305$ GHz, indicating a coupling rate between the transmons of $g/2\pi = 560$ MHz (Fig. 1c).

To determine the coupling rates of the modes into the waveguides, we measure the reflection coefficient, $r$, from each waveguide as a function of frequency and power, and globally fit a model based on a Lindblad master equation and input-output theory to the data[52] (see "Methods"). We determine the radiative coupling rates of the symmetric (antisymmetric) mode to the respective waveguide to be $\Gamma_s/2\pi = 2.87$ MHz ($\Gamma_a/2\pi = 2.83$ MHz). At the same time, we find additional coupling rates of $\Gamma_s'/2\pi = 98$ kHz ($\Gamma_a'/2\pi = 97$ kHz) to other channels, which we ascribe to unintended coupling of each mode to the opposite waveguide. Notably, the transition $|0\rangle \leftrightarrow |s\rangle$ ($|0\rangle \leftrightarrow |a\rangle$) exhibits overcoupling to waveguide S (A) with a selectivity ratio of $\Gamma_s/\Gamma_s' = 29$ ($\Gamma_a/\Gamma_a' = 29$).

### Noise-assisted excitation transport
We first study the impact of dephasing noise on the $|0\rangle \leftrightarrow |s\rangle$ and $|0\rangle \leftrightarrow |a\rangle$ transitions. To do so, we apply filtered white noise to the flux line connected to qubit 1, with a flat spectral profile $S_\Phi(\omega)$

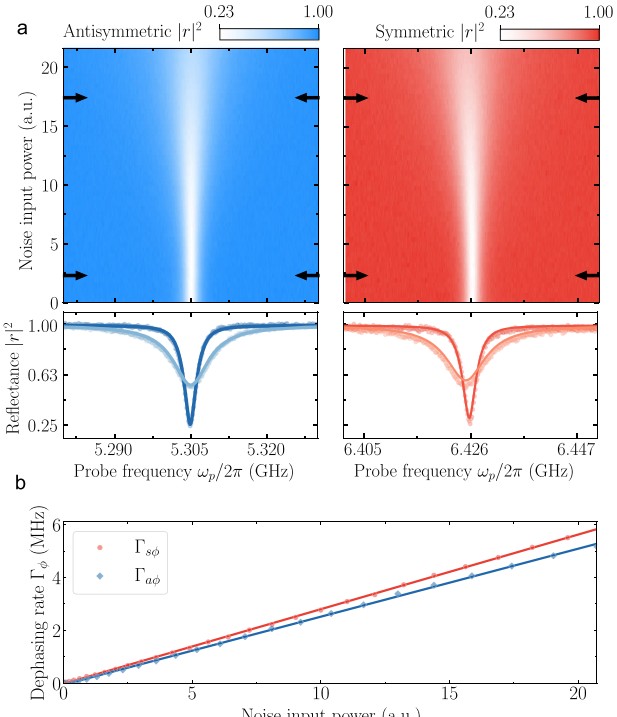

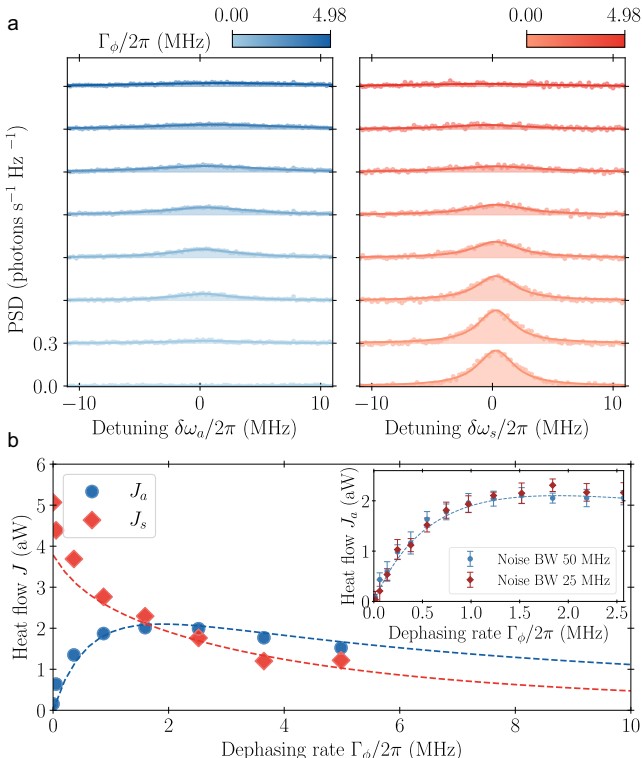

**Fig. 2 | Dephasing rate characterization. a** Top: 2D plot of power reflectance, $|r|^2$, as a function of probe frequency, $\omega_p$ and injected noise power for the antisymmetric (left, blue) and symmetric (right, red) mode. Bottom: Line cuts of reflectance at selected noise powers indicated by the arrows in the top panels. Solid lines are fits based on our theoretical model (see "Methods"). **b** Dephasing rates $\Gamma_{s\phi}$ and $\Gamma_{a\phi}$ of symmetric and antisymmetric mode, respectively, against input noise power, as extracted from the data in (**a**).

**Fig. 3 | Power transfer.** Measured power transfer between waveguides for coherent excitation of the symmetric mode at the Rabi frequency $\Omega_s/2\pi = 1.47$ MHz. **a** Power spectral density (PSD) extracted simultaneously from both the antisymmetric (blue) and symmetric (red) waveguide for different dephasing rates. Contributions from elastic scattering have been subtracted from the data. **b** Heat flow obtained from the integrated PSD from both waveguides as a function of dephasing. The inset highlights the power into the antisymmetric waveguide under two distinct noise bandwidths. The error bars show the variance of 49 measurements, each with 1 million averages. Dashed lines represent the result of master equation simulations (see "Methods").

characterized by a 50 MHz bandwidth centered at $2g$ and tunable amplitude. For increasing noise power, the reflectance exhibits broadening of the linewidth in both the antisymmetric and symmetric mode (Fig. 2a). Employing the same theoretical framework used for determining the coupling rates, the dephasing rate for varying noise power can be determined from a global fit with only the frequency and $\Gamma_\phi$ as variable parameters (Fig. 2b). The observed linewidth broadening primarily stems from the frequency components of the applied noise spectrum $S_\Phi(\omega)$ that bridges the energy gap between state $|s\rangle$ and $|a\rangle$. We ascertain this from the observation that substantial broadening is only noticeable when the spectral profile of the flux noise overlaps in frequency with $2g$. Furthermore, upon varying the bandwidth of the noise for a fixed amplitude, the linewidth saturates when the noise bandwidth exceeds the mode linewidth $\Gamma_{\{s,\,a\}}$ (see Supplemental Material[53]).

We then demonstrate dephasing-mediated energy transfer between waveguides by coherently exciting the symmetric state $|s\rangle$ through waveguide S using a continuous tone at the frequency $\omega_s$. With increasing drive strength, we observe a Mollow triplet obtained by driving the symmetric state towards saturation[54], which we use as a reference to calibrate subsequent PSD measurements from waveguide S, $S_s(\omega)$ (see "Methods"). Similarly, we calibrate the PSD from waveguide A, $S_a(\omega)$. Driving only through waveguide S at an amplitude corresponding to a Rabi frequency $\Omega_s/2\pi = 1.47$ MHz, we simultaneously measure both $S_s(\omega)$ and $S_a(\omega)$ as a function of the dephasing rate (Fig. 3a). In the absence of dephasing, no photons are detected through waveguide A, while $S_s(\omega)$ displays a broad resonance fluorescence spectrum, indicative of inelastically scattered photons against a two-level system [bottom orange line in Fig. 3a]. The emitted power from the two modes into their respective waveguides is obtained by integrating the measured PSD. For increased dephasing rates, the total

power re-emitted from the symmetric mode decreases monotonically (Fig. 3b). By contrast, the power detected in waveguide A initially rises sharply to a pronounced maximum before it starts to decrease. The initial rise results from the system overcoming its energy mismatch through noise-induced incoherent transitions between the symmetric and antisymmetric states. We again verify that it is the frequency components of the noise that bridge the energy gap $2g$ that enables excitation transport by reducing the bandwidth of the noise, for which no change in power transfer is observed (inset in Fig. 3b). For increasing dephasing rates, the power transfer is suppressed because of population localization, referred to as the quantum Zeno effect, a known feature of noise-assisted transport[47,50].

We explain the observed power transfer by a model based on the Lindblad master equation[55] (see "Methods"). In the presence of dephasing of qubit 1, the system exhibits longitudinal coupling through the $\sigma_z^1$ operator to the dephasing current in the flux line, characterized by the noise spectral density $S_\Phi(\omega)$. Expressed in the symmetric-antisymmetric basis this coupling takes the form $\frac{1}{2}(\sigma_z^s + \sigma_z^a) + \sigma_s^+\sigma_a^- + \sigma_a^+\sigma_s^-$. The initial two terms represent collective dephasing of the $|s\rangle$, $|a\rangle$ subspace, influenced by the zero-frequency component $S_\Phi(0)$ of the noise. Conversely, the subsequent cross terms enable interaction between the two modes, harnessing the frequency components at $S_\Phi(2g)$ to connect the two states, thereby enabling power transfer. Based on this model and on independently obtained estimates of the system parameters, we calculate heat flows in excellent agreement with the measured ones [dashed lines in Fig. 3b].

## Demonstration of noise-powered refrigeration

Having specified the setup of our device and its capabilities for energy transport, we demonstrate its capacity to refrigerate propagating microwave modes. We regulate the temperature of the heat baths using room-temperature electronics that generate radiation with a white-noise spectral profile over a limited frequency range. This radiation is transmitted through microwave coaxial cables connected by dissipative microwave attenuators, which are thermalized at different temperature stages of the cryostat. These attenuators serve a dual purpose; they attenuate the radiation's power and introduce quantum thermal noise[56,57]. At the base temperature of 10 mK, the final attenuator predominantly contributes quantum vacuum noise, ensuring the quantum nature of the fields. The radiation then reaches the symmetric and antisymmetric waveguides, with its bandwidth deliberately chosen to encompass the transitions $|0\rangle \rightarrow |s\rangle$ and $|0\rangle \rightarrow |a\rangle$ respectively. Within this frequency range, the radiation approximates a thermal field. However, outside the range, the radiation's spectral characteristics diverge from those of conventional black-body radiation. To reflect this distinction, we refer to this type of radiation as quasithermal. The interaction between the baths and the system is governed by the local spectral density of the radiation at the relevant transition frequencies, with the effective temperature $T$ determined by the average photon number $n = 1/[\exp(\hbar\omega/k_BT) - 1]$. By adjusting the power of the synthesized noise, we control the bath population, thereby regulating the population of the two-level systems. The photon occupation of the baths at specific noise power levels is characterized by measuring their PSD.

We keep a constant dephasing rate $\Gamma_\phi/2\pi = 0.94$ MHz and the temperature of the symmetric waveguide at $T_s = 177$ mK. We linearly increase the antisymmetric bath temperature from $T_a = 39$ mK (the minimum temperature we achieve in our waveguide[40]) to $T_a = 217$ mK, varying the temperature ratio $T_a/T_s$ in the range of 0.22 to 1.23, and measure the corresponding heat flows into each waveguide (Fig. 4a). To enhance measurement efficiency, we utilize instantaneous power measurements (see "Methods"), which are calibrated against a single integrated PSD measurement at a specific temperature ratio. For most temperatures, we measure heat flowing into the colder waveguide and out of the hotter one. However, our measurements indicate a temperature region in which the heat flows are reversed compared to the temperature gradient, signaling energy flows analogous to refrigeration [shaded region in Fig. 4a].

To elucidate the mechanism behind heat transport in our system, consider the interplay between separate thermalization to the baths and the effect of dephasing. Without dephasing, the symmetric (antisymmetric) transition thermalizes to waveguide S (A). The steady-state occupations of states $|a\rangle$ and $|s\rangle$ are, therefore, $P_{\{a, s\}} \approx n_{\{a, s\}}$, where we have taken the limit $n_{\{a, s\}} \ll 1$ for simplicity of discussion. The dephasing noise acts like an infinite-temperature bath for the $\{|s\rangle, |a\rangle\}$ subsystem, driving it towards an equal mixture of $|s\rangle$ and $|a\rangle$. Depending on the initial populations before dephasing-induced mixing, the modes either absorb or emit photons to/from their respective baths to reestablish thermal equilibrium, evidenced by either a peak or a dip in their PSD (Fig. 4b). This interplay between separate thermalization and population balancing due to dephasing manifests as heat flow. For the system to operate as a refrigerator, with the antisymmetric mode linked to the colder reservoir $T_a < T_s$, the initial population condition must satisfy $P_a > P_s$. Treating the modes as separate two-level systems, their average populations follow Fermi-Dirac statistics, leading to the refrigeration criterion $\omega_a/\omega_s < T_a/T_s < 1$. For our device, $\omega_a/\omega_s = 0.83$, which remarkably coincides, within experimental uncertainty, with the observed threshold value at which the sign of the heat currents is reversed and the device transitions into refrigeration mode.

Analytical calculations of the heat flows based on the Lindblad master equation[55] quantitatively reproduce the observed heat flows [blue and red lines in Fig. 4a]. Based on the theoretical model, we also

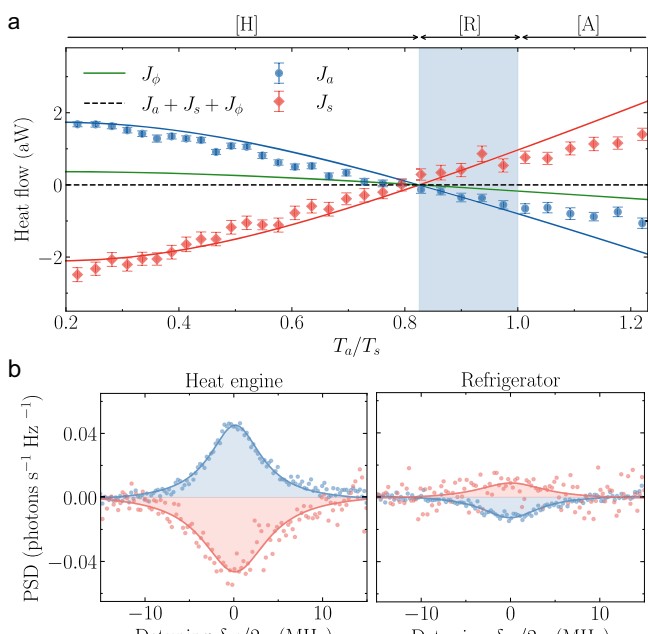

**Fig. 4 | Temperature-dependent operational modes of the device. a** Heat flows from instantaneous power measurements through the antisymmetric (blue) and symmetric (red) waveguides as a function of the temperature ratio $T_a/T_s$ at a fixed dephasing rate $\Gamma_\phi = 0.94$ MHz. Whilst $T_s$ remains fixed at 177 mK, $T_a$ is increased from base temperature to 217 mK. The first region [H] marks the operational regime of a heat engine, [R] that of a refrigerator (shaded blue region) and [A] a thermal accelerator. The error bars show the standard deviation of 63 measurements, each with 1 million averages. Solid lines are theoretical predictions from our analytical expression of the heat flows through the two waveguides and the dephasing channel (green solid line) used with independently extracted parameters. The dashed line indicate the sum of the theoretical predictions $J_a + J_s + J_\phi$. **b** Differential power spectral density (PSD) measurement in the heat engine operational regime [H] at $T_a/T_s = 0.2$ and in the center of the refrigeration region [R] at $T_a/T_s = 0.92$.

calculate the heat flow through the dephasing channel $J_\phi$ (green line). Considering all the heat flows, three distinct operational regions emerge as the ratio $T_a/T_s$ increases. In the region where $0 < T_a/T_s < 0.83$, the system exhibits energy flows characteristic of a heat engine [H] with $J_s < 0$ and $J_a, J_\phi > 0$ as defined in Reference[58]. In the interval $0.83 < T_a/T_s < 1$, the antisymmetric bath temperature remains cooler than the symmetric bath and the heat flows are $J_a, J_\phi < 0$ and $J_s > 0$. In this configuration, the dephasing channel supplies the necessary work to drive the heat flow against the temperature gradient and the system functions as a refrigerator [R]. As the temperature ratio surpasses $T_a/T_s > 1$, the formerly colder bath becomes the hotter one, leading to heat flowing from bath A (now hot) to bath S (now cold). In contrast to the heat engine, which exhibits similar energy flows, energy input from the dephasing channel is required due to $\omega_a$ being smaller than $\omega_s$. Consequently, this regime aligns with the operational mode of a thermal accelerator [A], characterized by heat flows $J_a, J_\phi < 0$ and $J_s > 0$. In all regions, adherence to the fundamental laws of thermodynamics is observed, when interpreted within the effective model of the noise-assisted three-level quantum thermal machine. For the first law $J_a + J_s + J_\phi = 0$ and for the second law $\dot{\Sigma} = \frac{J_a}{T_a} + \frac{J_s}{T_s} + \frac{J_\phi}{T_\phi} \geq 0$, where $\dot{\Sigma}$ denotes the effective entropy production rate and $T_\phi$ represents the temperature of an infinite temperature bath. A key metric for assessing a refrigerator's performance is its coefficient of performance $COP = \frac{J_a}{J_s - J_a}$, which gauges the efficiency of heat transfer from the cold to the hot reservoir[28,59]. We achieve a constant COP of 4.68 within the refrigeration regime, upper bounded by the Carnot limit $COP_{Carnot} = \frac{T_a}{T_s - T_a} = 4.88$ obtained from the temperatures at the crossover point between the heat engine and refrigerator. A significant

factor influencing the efficiency of the refrigerator is the coupling rate $g$ between the transmons. Theoretical modeling shows a sharp decrease in COP going from smaller to larger coupling rates with a convergent behavior towards $COP_{Carnot}$ for larger $g$ (see Supplemental Material[53]).

## Discussion

In a proof-of-principle experiment, we have realized a system that functions as a noise-assisted three-level quantum refrigerator, capturing the distinctive energy-flow dynamics of the model. We demonstrated how the interplay between individual thermalization and noise-assisted excitation transport can facilitate energy flow in steady state, both with and against an effective temperature gradient. With precise control over device parameters, our machine displays the heat flow characteristics of a heat engine, thermal accelerator, and refrigerator. Because the dephasing noise effectively acts as an infinite-temperature resource, the steady-state currents are equivalent to those of a minimal quantum absorption refrigerator[39], in which the three transitions of a three-level system each couple to a different bath.

The use of baths with spectrally localized heating justifies modeling the system with this simplified three-level structure because transitions to higher-excitation manifolds are not thermally activated. In our implementation, the three levels correspond to the ground state and the first excited symmetric and antisymmetric states of the artificial molecule. The two microwave waveguides realize the cold and hot baths by selectively coupling to the corresponding transitions. Finally, the engineered dephasing channel acts as an effective infinite-temperature bath by inducing incoherent transitions between the two excited states. When the device operates as a refrigerator, the noise provides the energy quanta needed to bridge the energy gap between the symmetric and antisymmetric levels. In this way, it enables population transfer against the thermal gradient and thereby implements the work input in the absorption-refrigerator model. This simplification leads to analytical expressions for the heat flows within the framework of the Lindblad master equation, which is the de facto tool to study quantum thermodynamics[60]. In this Lindbladian approach, thermal noise is assumed to be locally flat and white. Consequently, an idealized quantum system driven by true thermal noise and one driven by locally white noise would exhibit, within this framework, the same dynamics. The quantitative agreement between theory and measured heat flows, without fitted free parameters, suggests that the local white-noise approximation replicates the essential dynamics observed in our system.

Although our method utilizes quasithermal baths, the setup is adaptable to natural thermal sources. Possible examples include the thermal radiation from various temperature stages of a dilution refrigerator[61] or a heated resistor[62], preferably in combination with infrared-blocking filters to prevent the generation of quasiparticles in the superconducting circuit[63]. Even when a true thermal field is sourced, specific transitions could still be targeted through the use of bandpass filters[62]. In our present implementation, the hot and cold reservoirs are limited-bandwidth propagating microwave fields with controllable photon occupation. The observed refrigeration therefore corresponds to a reduction in the mean photon number of the cold reservoir modes. To connect with a more conventional setting, such modes could be strongly coupled to finite-heat-capacity solids, such as 50 $\Omega$ resistors, in which case the extracted heat should lead to a measurable temperature drop of the cold object. For this reason, and given the engineered nature of the reservoirs, we frame our results as an analog quantum simulation[64] of a noise-assisted refrigerator.

Our work establishes a technique to measure tiny heat flows in microwave waveguides, at the aW level and below, using a linear amplification chain. The sensitivity of these measurements could be further improved by incorporating a nearly quantum-limited amplifier in the chain. A compelling next step, which would benefit from such an

improvement, is the measurement of fluctuations in the heat flows. The ability to measure these fluctuations in real time will facilitate investigations in the thermodynamics of precision, including the extension of thermodynamic uncertainty relations (TUR) to the quantum domain[12] and a direct measurement of the energy costs of timekeeping[18,19].

## Methods

### Reflection spectroscopy

The reflection coefficient $r$ for a two-level system coupled to the end of a waveguide has been derived using a Lindblad master equation and input-output theory[57,65]. In our device, the lowest states of the single-excitation manifold, corresponding to the symmetric and antisymmetric modes, can in the zero-temperature limit, each be treated as an effective two-level system. The reflection coefficient for each mode $i = \{s, a\}$ is then given by:

$$r(\omega - \omega_i) = 1 - \frac{i\Gamma_i\Gamma_{1i}(\omega - \omega_i - i\Gamma_{2i})}{\Omega_i^2\Gamma_{2i} + \Gamma_{1i}\left[(\omega - \omega_i)^2 + \Gamma_{2i}^2\right]}. \tag{2}$$

Here, $\Gamma_i$ denotes the coupling rate between mode $i$ and its corresponding waveguide, while $\Gamma_i'$ is the coupling rate to all other decay channels. These define the total energy relaxation rate $\Gamma_{1i} = \Gamma_i + \Gamma_i'$ and the total decoherence rate $\Gamma_{2i} = (\Gamma_i + \Gamma_i')/2 + \Gamma_{i\phi}$, where $\Gamma_{i\phi}$ is the pure dephasing rate. The parameter $\Omega_i = 2\sqrt{\frac{P_{in}A\Gamma_i}{\hbar\omega_i}}$ is the Rabi frequency, relating the drive strength to the input power $P_{in}$ of the coherent tone and the total attenuation $A$ of the input line. Assuming negligible dephasing, the coupling rates and attenuation are obtained from circular fits of the complex reflection coefficient as a function of input power around the mode frequencies $\omega_{s,a}$, (see example shown in Fig. 5). At low power, the data trace out a near-unit circle in the IQ plane, collapsing towards a single point as the power is increased. This behavior reflects the crossover from predominantly coherent to incoherent scattering as the two-level system approaches saturation[57].

### Power spectral density calibration

We determine the effective temperatures of the modes in the microwave waveguides, when populated with synthesized quasithermal radiation, from direct measurements of their power spectral density (PSD). To accurately infer the power spectrum of the bosonic environment in the waveguides before the amplification chain, we first establish a calibrated reference signal. We perform this calibration by measuring the spectra $S_s(\omega)$ and $S_a(\omega)$ while coherently driving the symmetric and antisymmetric transitions respectively. At sufficiently high drive power, we observe a fully resolved Mollow triplet[54] for both

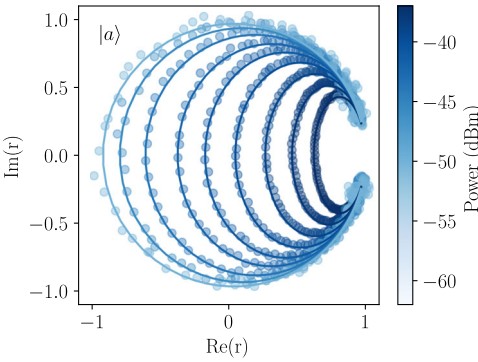

**Fig. 5 | Rate characterization.** Typical complex reflection coefficient $r$ in the IQ plane for the antisymmetric ($|a\rangle$) mode as a function of input power. Points show the measured data and solid curves the fitted response of a driven two-level system. Fitted reflection data is used to extract the coupling rates as well as the total attenuation of the input lines.

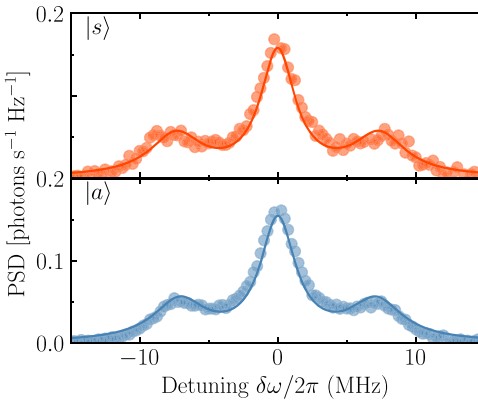

**Fig. 6 | Power spectral density.** Resonance-fluorescence spectra (Mollow triplets) of the symmetric ($|s\rangle$) and antisymmetric ($|a\rangle$) modes under strong resonant drive. Points show the measured power spectral density and solid lines the corresponding fits.

modes (see Fig. 6), featuring distinctly resolved side peaks. Subtracting the measured spectral profile of the background, we fit the well known Mollow triplet expression to the resonance fluorescence spectrum[66]. In the limit of strong drive, $\Omega \gg \Gamma_{s,a}$, this expression reduces to

$$
S_i(\omega) \approx \frac{1}{2\pi} \frac{\hbar\omega\Gamma_i}{4} \left\{ \frac{\Gamma_n}{(\delta\omega_i + \Omega)^2 + \Gamma_n^2} + \frac{2\Gamma_{2i}}{\delta\omega_i^2 + \Gamma_{2i}^2} \right.
$$
$$
\left. + \frac{\Gamma_n}{(\delta\omega_i - \Omega)^2 + \Gamma_n^2} \right\}. \tag{3}
$$

Here $\delta\omega_i$ is the detuning frequency and $\Gamma_n = (\Gamma_{1i} + \Gamma_{2i})/2$. According to this calibration, we scale the subsequent PSD measurements by a factor obtained as the ratio between the theoretical height of the center peak and the measured data.

## Heat flow measurements

We determine the effective temperatures of the synthesized thermal baths as a function of the applied noise power by using the calibrated power spectral density (PSD). For a fixed bath temperature, the heat currents of interest constitute only a small fraction of the total output power, referred back to the device input (i.e., before the amplification chain). Using Parseval's theorem, we estimate this power from time-domain records of the complex field signals $f_{s,a}(t)$ by measuring the averaged integrated energy

$$
\langle Q_{s,a} \rangle = \left\langle \int_0^T |f_{s,a}(t)|^2 dt \right\rangle \tag{4}
$$

with an integration window $T = 4\,\mu s$. To isolate the contribution of the device from the background, we perform interleaved measurements with the thermal machine in an "on" configuration (dephasing applied) and an "off" configuration (no dephasing). We then obtain the net heat current as

$$
J_{s,a} = \frac{1}{T}\left(\langle Q_{s,a}^{on}\rangle - \langle Q_{s,a}^{off}\rangle\right) \tag{5}
$$

A positive value of $J_{s,a}$ indicates that the device emits photons into the corresponding waveguide when activated, whereas a negative value signifies absorption. This method is computationally inexpensive; to calibrate it, we use the power obtained by integrating a known, properly scaled PSD, such as the power transfer shown in Fig. 3, as a reference.

## Theoretical model and numerical simulations

In the case of two ideally hybridized qubits with $\omega_1 = \omega_2 = \omega$, the Hamiltonian of the system is well approximated by Eq. (1). The dynamics relevant for heat-flow calculations between the reservoirs is described by the Lindblad master equation

$$
\dot\rho = \mathcal{L}\rho = -i[\mathcal{H}, \rho] + \mathcal{L}_s\rho + \mathcal{L}_a\rho + \frac{\Gamma_\phi}{2}\mathcal{D}[\sigma_s^+\sigma_a^- + \sigma_a^+\sigma_s^-]\rho. \tag{6}
$$

Here $\rho$ is the density matrix and $\mathcal{D}[X]$ is the Lindblad superoperator

$$
\mathcal{D}[X]\rho = X\rho X^\dagger - \frac{1}{2}X^\dagger X\rho - \frac{1}{2}\rho X^\dagger X. \tag{7}
$$

The terms $\mathcal{L}_s$ and $\mathcal{L}_a$ contain the dissipation taking place between the waveguides populated with $n_j$ photons, and their respective state transition

$$
\mathcal{L}_{j=\{s,a\}}\rho = \Gamma_j(n_j + 1)\mathcal{D}[\sigma_j^-]\rho + \Gamma_j n_j \mathcal{D}[\sigma_j^+]\rho. \tag{8}
$$

In the steady state, the average energy of the system is constant, so the net energy flow through all channels must vanish[55],

$$
\frac{d}{dt}\langle\mathcal{H}\rangle = \text{Tr}(\mathcal{H}\mathcal{L}\rho) = 0. \tag{9}
$$

We thereby calculate the individual heat flows into each waveguide as:

$$
J_{j=\{s,a\}} = \text{Tr}\left(\mathcal{H}\mathcal{L}_j\rho\right)
$$
$$
\approx \frac{\hbar(n_a - n_s)\Gamma_a\Gamma_s\Gamma_\phi(g \pm \omega)}{\Gamma_s\Gamma_\phi + \Gamma_a(2\Gamma_s + \Gamma_\phi)} \tag{10}
$$

where we obtained the approximate expression via a first-order expansion in $n_s$ and $n_a$. Similarly, we obtain the heat flow into the dephasing channel as:

$$
J_\phi = \text{Tr}\left(\mathcal{H}\frac{\Gamma_\phi}{2}\mathcal{D}[\sigma_s^+\sigma_a^- + \sigma_a^+\sigma_s^-]\rho\right)
$$
$$
\approx -\frac{2g\hbar(n_a - n_s)\Gamma_a\Gamma_s\Gamma_\phi}{\Gamma_s\Gamma_\phi + \Gamma_a(2\Gamma_s + \Gamma_\phi)}. \tag{11}
$$

Remarkably, we find that the heat flows predicted by Eq. (10) are in excellent agreement with the experimental data shown in Fig. 4, using only independently extracted system parameters and no free fit parameters.

To calculate the power spectrum at the symmetric mode in the presence of a coherent drive with frequency $\omega_d = \omega_s$, we again solve the Lindblad master equation, but with the Hamiltonian;

$$
\mathcal{H} = \sum_{i=1,2}(\omega_i - \omega_d)\sigma_i^+\sigma_i^- + g(\sigma_1^+\sigma_2^- + \sigma_2^+\sigma_1^-) + \frac{\Omega_s}{2}(\sigma_s^+ + \sigma_s^-). \tag{12}
$$

Numerically solving the master equation with this Hamiltonian, the heat flow into the antisymmetric waveguide is found as $J_a = \text{Tr}(H\mathcal{L}_a\rho)$. The incoherent emission from the symmetric side is described by its power spectral density

$$
S_s(\omega) = \Gamma_s \int_{-\infty}^{\infty} \langle\sigma_s^+(\tau)\sigma_s^-(0)\rangle e^{-i\omega\tau} d\tau. \tag{13}
$$

which we integrate to obtain the theoretical predictions seen in Fig. 3.

Further details can be found in the Supplemental Material[53].

## Data availability

Supporting data are available in the figshare data repository https://doi.org/10.6084/m9.figshare.30693494.

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

## Acknowledgements

The presented device design was assisted by the Python package QuCAT[67] and was fabricated in Myfab Chalmers, a nanofabrication laboratory. This work received support from the Swedish Research Council via Grant No. 2021-05624, the Knut and Alice Wallenberg Foundation through the Wallenberg Center for Quantum Technology (WACQT), from the European Research Council via Grant No. 101041744 ESQuAT, and from the European Union via Grant No. 101080167 ASPECTS.

## Author contributions

S.G. conceived the original idea for the project. M.A.A. designed the device, which was then fabricated by C.M. S.S. conducted the experiments, assisted by V.M. and M.A.A. S.S. performed the theoretical analysis, with assistance from V.M. S.S. wrote the manuscript, with feedback from M.A.A. and S.G. M.A.A. and S.G. supervised the project. All authors contributed to the discussions and interpretation of the results.

## Funding

## Competing interests

S.G. is a co-founder and equity holder in Sweden Quantum AB. The remaining authors S.S., M.A.A., C.M., and V.M. declare no competing interests.
