## [Transparent Peer Review file · Nature Communications]

Quantum refrigeration powered by noise in a superconducting circuit

Corresponding Author: Mr Simon Sundelin

Version 0:

Reviewer comments:

Reviewer #1

(Remarks to the Author)

The manuscript of Sundelin et al. presents careful measurements of a two-qubit system on a superconducting platform. Artificial noise with a particular spectrum around different resonances is created at room temperature, and power spectral densities on two waveguides are measured to determine effective “heat currents”.

I find this manuscript not to present what it claims to do. The narrative circles around phrases like “synthesized thermal fields”, “bring closure to a long-standing quest for a Brownian refrigerator”, “thermally populated waveguides” already in abstract. The artificial “noise” generated in a narrow band around the expected transitions is very far from true thermal noise, though it certainly leads to corresponding transitions according to Fermi’s golden rule. One is left with an impression that the experiment aims to mimic thermodynamics, but there is no clue that the dynamics of the system would be the same with true thermal noise, which would be the real test of such a system but remains elusive after this work.

Besides the abstract, the manuscript puts forward many overstatements or misleading phrases such as “The refrigerator is driven by thermal fluctuations” and “regulate the temperature of the baths by employing synthesized thermal fields”. I do not agree with the following statement “In our experiment, the thermal baths are composed of classically synthesized microwave noise with finite bandwidth”. This is far-fetched, since a thermal bath is an equilibrium (infinite) reservoir which emanates thermal noise.

The manuscript should be rewritten, removing the artificial connection to thermodynamics and the claim of closing the long-standing quest for a Brownian refrigerator. I believe, on the other hand, it describes a neat experiment on a controlled two-qubit system.

Reviewer #2

(Remarks to the Author)

The most important result in this article presents an experimental realization of a novel Brownian-type quantum refrigerator using superconducting circuits. The authors demonstrate a novel quantum thermal machine that leverages noise-assisted quantum transport to facilitate cooling in steady state. Specifically, the authors implemented a quantum refrigerator using two coupled transmon qubits. The design, which exploits symmetry-selective couplings between the qubits and two microwave waveguides, is both elegant and effective. The use of synthesized thermal fields to regulate waveguide temperatures and the ability to measure aW power transport, demonstrate excellent control over the engineered quantum system. This experiment utilizes a similar engineered system from authors’ previous work in Ref. 52 for frequency conversion of microwave photons and was envisioned to work for selective heat transport.

The work, data collection, and analysis are of high quality and though niche, there is a level of novelty in the experiment that warrants the publication Nature Communications. Here are some comments that would help to improve the manuscript:

1. Since Nature Communications have a broad audience, could the authors comment if aW heat transport measurement sensitivity is state-of-the-art, and what are the limiting factors to better sensitivity?

2. The coefficient of performance (COP) is highlighted as a key metric. As a reader, I would like to know what is the COP especially in the refrigeration regime. Could this be plotted, for example, in Fig. 4 (a)?

Version 1:

Reviewer comments:

Reviewer #1

(Remarks to the Author)

I have read the new version of the manuscript, and especially the response letter of the authors very carefully. Regrettably I do not see a reason to change my initial recommendation not to publish this work. My negative recommendation is based on the two main concerns: 1. the use of artificial "white" noise synthesized by a waveform generator over a selected narrow band, and 2. lack of true thermal baths. The new version and the response letter of the authors did not convince me further on these points.

I am also surprised that based on my criticism on the two main points above, raised in my previous report already, the authors further strengthen their main general claim in the abstract in the revised manuscript, from "bring closure to long-standing quest of a Brownian refrigerator" into "is the first demonstration of a Brownian refrigerator"! To be fair, this statement should be removed if the work is going to be published somewhere.

I regret that I cannot recommend publishing this article on a well-conducted experiment because of its overblown claims.

Reviewer #2

(Remarks to the Author)

I have now read the revision from the authors and now satisfied with the latest revision, thus recommend publication.

Reviewer #3

(Remarks to the Author)

The result is seemingly very relevant, but I tend to agree with Referee 1 in that the presentation is overstating the result and partly misleading.

The manuscript purportedly demonstrates "a novel quantum thermal machine that leverages noise-assisted quantum transport to enable a cooling engine in steady state". But, as far as I understand, no physical element is being refrigerated, so this circuit is not really a cooling engine.

Instead, the authors seem to demonstrate energy flows under very specific conditions that would correspond with refrigeration if no further degrees of freedom would exist in the system. This is a very relevant result, but I would rather consider it to fall within the context of quantum analog simulators. The noise-assisted three-level refrigerator is a known toy model in the quantum thermodynamics literature and this is possibly the first experimental implementation approaching the model.

It is not clear at all to me how negative powers are calculated, and this is a crucial aspect of the results. The supplementary material states:

„The power is directly obtained as the integral of the recorded time trace $f(t)$ squared from each waveguide, in accordance with Parseval's theorem. This is computationally inexpensive and is calibrated against the integral of a known PSD measurement such as the Mollow triplet.“

How can the sum of absolute values squared ever provide a negative number?

Also, after equation S1 the following statement reads: „At low power the reflectance goes towards a near unit circle in the IQ plane while reducing towards a single point for lower powers.“ Do you mean "for higher powers"?

Version 2:

Reviewer comments:

Reviewer #4

(Remarks to the Author)

I have read the revised manuscript, the authors' rebuttal, and the correspondence from the previous review rounds. I understand that my role is to adjudicate the manuscript, particularly in light of the concerns raised by Reviewer 1 and Reviewer 3 regarding the scope of the claims. The point was to reframe the work as an analogue quantum simulation of a noise-assisted refrigerator, rather than the demonstration of a physical cooling device.

In my assessment, the authors have mostly resolved the previous concerns. The revision to reframe the results as an analogue quantum simulation positions the work appropriately. By removing claims of demonstrating a "Brownian refrigerator" and clarifying that the observed effect is a reduction in the photon occupation of microwave modes rather than the cooling of a physical object, the manuscript is now clearer and more precise. I believe that the experimental work itself is of high quality, demonstrating control over the system and sensitivity in measuring attowatt-scale heat currents. While I have to admit that I am not really an expert in the field of quantum thermodynamics, the experimental demonstration of the core energy-flow dynamics of a noise-assisted three-level quantum refrigerator model seems to be a timely contribution to the field and also of broad interest, and I believe the manuscript is now well-suited for publication in Nature Communications.

The following suggestions are intended to further improve the presentation of the work.

1. In the Discussion section, the authors could briefly elaborate on the precise mapping between their experimental system (a superconducting circuit with engineered reservoirs) and the idealized theoretical model it simulates (a three-level system coupled to ideal thermal baths). A clearer statement on which features of the ideal model are captured and which are not would be helpful.
2. The manuscript identifies that the dephasing noise acts as a resource, analogous to an infinite-temperature bath, that enables the device to function. For readers less familiar with this specific model, it would be helpful to include a sentence or two clarifying why this incoherent channel can be interpreted as the "work" input in the context of an absorption refrigerator.
3. In the Discussion section, the sentence ending "...analogue quantum simulation [?] of a noise-assisted refrigerator" appears to have a placeholder for a citation.

We address the specific comments of the Reviewers point by point below, where we have quoted the Reviewer in **bold font** and replied in normal font.

Reply to Reviewer 1

The manuscript of Sundelin et al. presents careful measurements of a two-qubit system on a superconducting platform. Artificial noise with a particular spectrum around different resonances is created at room temperature, and power spectral densities on two waveguides are measured to determine effective “heat currents”.

We thank the Reviewer for going through the manuscript and appraising our work. We have considered all the issues raised and have addressed them below along with necessary modifications in the manuscript.

1) I find this manuscript not to present what it claims to do. The narrative circles around phrases like “synthesized thermal fields”, “bring closure to a long-standing quest for a Brownian refrigerator”, “thermally populated waveguides” already in abstract. The artificial “noise” generated in a narrow band around the expected transitions is very far from true thermal noise, though it certainly leads to corresponding transitions according to Fermi’s golden rule. One is left with an impression that the experiment aims to mimic thermodynamics, but there is no clue that the dynamics of the system would be the same with true thermal noise, which would be the real test of such a system but remains elusive after this work.

We thank the reviewer for their feedback and for encouraging a more detailed description of the physical heat baths.

Our synthesized fields need only be locally white in the relevant frequency range of the qubit transitions to reproduce the essential dynamics of a thermal bath. In the standard Lindblad master equation (LME) approach, thermal noise is assumed to be locally flat and white, in accordance with the Markov approximation. Consequently, a quantum system driven by true thermal noise and one driven by locally white noise exhibit the same LME dynamics. Passive filtering of a true thermal field would similarly allow us to selectively address the relevant qubit transition frequencies, thus achieving the same local spectral density as our engineered noise. The quantitative agreement between theory and measured heat flows, without fitted free parameters, strongly suggests that the local white-noise approximation accurately replicates the essential dynamics enabled by a genuine thermal reservoir. We now expand on the details of our engineered heat baths to put them into clearer context.

We utilize room-temperature electronics to synthesize radiation with a white noise spectral profile over a selected frequency range. This synthesized radiation is injected into microwave

coaxial cables connected via dissipative microwave attenuators, which are thermalized at different cryostat temperatures. These attenuators not only reduce the radiation’s power but also introduce quantum thermal noise. At 10 mK, the final attenuator primarily contributes quantum vacuum noise, ensuring the quantum nature of the fields that interact with the system. The radiation ultimately reaches the symmetric and antisymmetric waveguides, with its bandwidth specifically chosen to include the transition frequencies $|0\rangle \rightarrow |s\rangle$ and $|0\rangle \rightarrow |a\rangle$. Within this frequency range, the radiation effectively approximates a thermal field. Outside this range, however, its spectral profile deviates significantly from natural blackbody radiation. To address this deviation and align with terminology in thermodynamics, we now refer to this effective thermal field as “quasithermal”. The qubits’ interactions with the bath are determined by the local spectral density of the radiation at the relevant transition frequencies, with the effective temperature T depending on the average number of quasithermal photons $n = 1/[\exp(\hbar\omega/k_B T_{\text{HC}}) - 1]$ at these frequencies. By regulating the synthesized noise’s power, we can control the photon number, thereby heating the respective two-level systems.

The Lindblad master equation is a standard tool for describing the dynamics of open quantum systems concerned with describing quantum systems connected to dissipative baths typically at zero temperature (only quantum vacuum fluctuations). It is also the de facto tool in the context of quantum thermodynamics, which explores the concept of heat, work and temperature in the same setting in which the baths may be at a finite temperature associated with an Ohmic spectral density. For reference, please see Ref. [32] [M. T. Mitchison, *Contemporary Physics* 60, 164 (2019)], and the newly added reference: Anders and Vinjanampathy, *Contemporary Physics* 57, 545 (2016). This tool is also used to calculate quantities such as heat flow and entropy production. See newly added reference for a good example: V. Balachandran et.al., *Phys. Rev. E* 99, 032136 (2019). As a Markovian master equation that neglects any memory effects of the reservoirs and any system-reservoir correlations, it naturally models thermal fields as flat white noise around the transitions of the quantum thermal machine. Therefore, the synthesized fields targeting the relevant transitions interact with discrete energy levels of the quantum systems in an equivalent way as real thermal fields do. Please refer to C. W. Gardiner et.al., *Phys. Rev. E* 31, 3761 (1985) for further discussion about the approximation of thermal fields as white noise.

In the ideal scenario, the quantum thermal machine would have only few discrete energy levels relevant to the machine’s operation while it interacts with real thermal baths. Any additional energy levels that interact with the thermal baths could potentially limit the thermal performance of the machine. In fact, it is desirable to have the ability to engineer the energy spectra of both the quantum thermal machine and the heat baths so that the myriad abstract, theoretical models of quantum thermodynamics can be studied experimentally with good control. For example, a minimal realization of a quantum absorption refrigerator (see reference [37-39] in the manuscript) is a three-level system, where each of the three transitions are coupled to three different thermal baths. Our scheme of engineered physical heat baths is actually well-suited to studies in quantum thermodynamics.

On the other hand, a true thermal spectral profile could alternatively be sourced directly from a resistor, with specific transitions targeted via bandpass filters as was done in Nature

Communication **13**, 1552 (2022). This approach, however, would require a different experimental setup, which we discuss in the outlook of the main text near the end. We acknowledge the reviewer’s suggestion to explore the dynamics of the system with true thermal noise or even alternative baths with different spectral distributions or photon statistics. This could shed light upon any inherent limitations of the Lindbladian description of quantum thermodynamics originating in the approximations behind its derivation. While our measurements of the first-order moment of the photon currents (heat currents) agree with predictions from the Lindbladian Master equation, higher-order moment such as current fluctuations is being addressed in our lab in a follow-up experiment currently underway.

Revisions

We have addressed the referee’s concerns in multiple ways throughout the manuscript:

1. Replaced the term “thermal” with “quasithermal”. including in terms such as “thermal photons”, “thermal reservoirs” and “thermal fields”. In some cases “heat baths” has been used instead. The manuscript now also conveys that the baths are associated with an “effective temperature”.
2. Revised the entire paragraph that describes the engineered physical heat baths. The distinction from a true thermal field is highlighted and we elaborate upon the quantum nature of the heat baths.
3. In the introduction, when mentioning the nature of our heat baths, we have included a now published reference of when this type of noise has been used as heat baths in a similar context, namely for a superconducting quantum absorption refrigerator “Thermally driven quantum refrigerator autonomously resets a superconducting qubit” Nat. Phys. (2025).
4. Added reference: V. Balachandran et.al., Phys. Rev. E **99**, 032136 (2019).
5. Added reference: Vinjanampathy and Anders, “Quantum Thermodynamics”, Contemporary Physics **57**, 545 (2016)
6. Added reference: Caves, “Quantum limits on noise in linear amplifiers,” Phys. Rev. D **26**, 1817 (1982).
7. Added a comment about the Lindblad master equation in the summary paragraph of the manuscript: “...Lindblad master equation, which is the de facto tool to study quantum thermodynamics”
8. Our stated description of a Brownian refrigerator as being driven solely by thermal fluctuations has been changed to follow the proper definition given in reference [42] of the manuscript; “It conveys heat unidirectionally in response to random noise”.

2) Besides the abstract, the manuscript puts forward many overstatements or misleading phrases such as “The refrigerator is driven by thermal fluctuations” and “regulate the temperature of the baths by employing synthesized thermal fields”.

As part of the answers and revisions to the previous point we have revised these statements to better adhere to their proper definitions. We hope that any statements that were previously found potentially misleading or seemingly overstating their importance have been avoided.

3) I do not agree with the following statement “In our experiment, the thermal baths are composed of classically synthesized microwave noise with finite bandwidth”. This is far-fetched, since a thermal bath is an equilibrium (infinite) reservoir which emanates thermal noise.

We agree that the usage of “thermal baths” in this context should be changed to better represent the “quasithermal” nature of our input fields. However, the semi-infinite waveguides that couple to our system supports a continuum of modes which at base temperature of the cryostat are populated by thermal photons and vacuum fluctuations. We further populate the modes which are spectrally localized around our targeted transition frequencies by injecting our engineered noise. Our baths can therefore be seen as (infinite) reservoirs. In addition, our engineered heat bath separates the incoming and outgoing fields using a microwave circulator. As a result, the effective temperature of the quasithermal bath is completely unaffected by the amount of energy exchanged with the quantum system.

Revisions

The terms “thermal baths” have been changed to “heat baths”.

4) The manuscript should be rewritten, removing the artificial connection to thermodynamics and the claim of closing the long-standing quest for a Brownian refrigerator. I believe, on the other hand, it describes a neat experiment on a controlled two-qubit system.

We understand that the Reviewer’s underlying concern stems from the fact that the utilized heat baths are not sourced from a true thermal source, but are instead composed of synthesized noise. In our response above, we have clarified the distinction between a proper thermal reservoir and our baths. We have elaborated on the nature of the bath in the manuscript. We have also presented an argument above of why our scheme of using synthesized thermal baths is suitable for experiments with quantum thermodynamics, at least for the first-order moments of photon currents. This is validated by the predictions of the Lindblad master equation that is widely used in the quantum thermodynamics community [Another reference: Patrick Potts, Quantum Thermodynamics, arXiv: 2406.19206]. We are currently investigating the higher-order moments of photon currents experimentally in a similar device.

After the revisions, including the more detailed description of the synthesized quasithermal fields, we hope that the direct connection to thermodynamics has been made clearer. Additionally, it also holds true that this—to our knowledge—is the first demonstration of a noise-powered (or Brownian) quantum refrigerator. We have, however, rephrased this now in the manuscript, including in the abstract.

We are glad to hear that the Reviewer is satisfied with our experiment. We thank them for prompting us to review the interpretation of results. We have considered the comments raised and have addressed them accordingly.

Revisions

We have rephrased “bring closure to long-standing quest for a Brownian refrigerator” to “is the first demonstration of a Brownian refrigerator”.

Reply to Reviewer 2

The most important result in this article presents an experimental realization of a novel Brownian-type quantum refrigerator using superconducting circuits. The authors demonstrate a novel quantum thermal machine that leverages noise-assisted quantum transport to facilitate cooling in steady state. Specifically, the authors implemented a quantum refrigerator using two coupled transmon qubits. The design, which exploits symmetry-selective couplings between the qubits and two microwave waveguides, is both elegant and effective. The use of synthesized thermal fields to regulate waveguide temperatures and the ability to measure aW power transport, demonstrate excellent control over the engineered quantum system. This experiment utilizes a similar engineered system from authors’ previous work in Ref. 52 for frequency conversion of microwave photons and was envisioned to work for selective heat transport.

The work, data collection, and analysis are of high quality and though niche, there is a level of novelty in the experiment that warrants the publication Nature Communications. Here are some comments that would help to improve the manuscript

We are glad to hear that the Reviewer finds our work appealing and of high quality. We have considered the comments raised and have addressed them below along with necessary modifications in the manuscript.

1) Since Nature Communications have a broad audience, could the authors comment if aW heat transport measurement sensitivity is state-of-the-art, and what are the limiting factors to better sensitivity?

We thank the reviewer for the question. The measurement of aW powers in microwave photon currents are indeed state-of-the-art. The limiting factor for better sensitivity boils down to quantum efficiency. Regardless of the quality of an amplification chain, it will always add thermal noise photons, dictated by Friis formula for added noise photons. This formula states that the largest contributing noise source in an amplification chain is the first amplifier. In this work we use a High Electron Mobility Transistor (HEMT) amplifier as the first low noise amplifier of our output line. The total noise contribution in these types of

setups is on the order of 15-20 photons, corresponding to a quantum efficiency of around 5 %. However, by utilizing a quantum limited amplifier such as a traveling wave parametric amplifier, we can achieve quantum efficiencies of up to 50 %. This increase could significantly increase the sensitivity of our measurements since the uncertainty scales quadratically with the quantum efficiency.

Revisions

We have added to the discussion section of the manuscript the potential for improved sensitivity by using a quantum-limited amplifier.

2) The coefficient of performance (COP) is highlighted as a key metric. As a reader, I would like to know what is the COP especially in the refrigeration regime. Could this be plotted, for example, in Fig. 4 (a)?

The coefficient of performance (COP) gauges the efficiency of heat transfer from the cold to the hot reservoir. From our theoretical model we extract the COP of 4.68 to be constant within the full refrigeration region and closest to the Carnot limit of 4.88 right at the crossover point from the heat engine to the refrigeration regime. After this crossover point the Carnot efficiency starts to deviate away from our COP. We have now highlighted this within the manuscript. However, because of the constant value and risk of making the figure too crowded we decided against including it in Figure 4 (a).

Revisions

We have clarified the section describing the COP.

We address the specific comments of the Reviewers point by point below, where we have quoted the Reviewer in **bold font** and replied in normal font.

Reply to Reviewer 1

I have read the new version of the manuscript, and especially the response letter of the authors very carefully. Regrettably I do not see a reason to change my initial recommendation not to publish this work. My negative recommendation is based on the two main concerns: 1. the use of artificial "white" noise synthesized by a waveform generator over a selected narrow band, and 2. lack of true thermal baths. The new version and the response letter of the authors did not convince me further on these points. I am also surprised that based on my criticism on the two main points above, raised in my previous report already, the authors further strengthen their main general claim in the abstract in the revised manuscript, from "bring closure to long-standing quest of a Brownian refrigerator" into "is the first demonstration of a Brownian refrigerator"! To be fair, this statement should be removed if the work is going to be published somewhere. I regret that I cannot recommend publishing this article on a well-conducted experiment because of its overblown claims.

We thank the reviewer for taking the time to carefully read the revised manuscript and our response letter. We sincerely regret that the previous revisions did not change the reviewer's assessment. In response and after additional feedback, we have carefully revised the manuscript to ensure that the presentation accurately reflects the intended scope and conceptual positioning of the work.

We fully acknowledge the reviewer's concern that our experiment does not employ true thermal reservoirs. Instead, we utilize spectrally localized white noise, synthesized via waveform generators, to emulate thermal fields with controllable effective temperatures. Furthermore, no physical subsystem is directly cooled; rather, the observed energy extraction pertains to spectrally defined propagating microwave modes — in particular, those coupled to the antisymmetric mode of the artificial molecule. The refrigeration we observe corresponds to a measurable reduction in the mean photon occupation of these modes. This situation is directly analogous to widely accepted cases in quantum thermodynamics and quantum optics, such as optomechanical sideband cooling, where a single mechanical mode — rather than a macroscopic solid — is cooled. In our case, the modes are propagating rather than confined, but the underlying principle of selectively reducing the occupation of specific field modes remains the same. The energy flow dynamics we measure are consistent with those expected from an idealized three-level quantum thermal machine, under the assumption that no additional degrees of freedom participate in the dynamics beyond the engineered reservoirs and the system itself.

In light of this, we have revised the presentation of the results in the manuscript and explic-

itly frame them as an analogue quantum simulation of a noise-assisted three-level quantum refrigerator. This framing accurately reflects that the experiment reproduces the predicted energy-flow dynamics of the model using engineered reservoirs with tunable spectral properties, rather than physical heat baths in the conventional thermodynamic sense. It also aligns our work with recent literature on analogue simulations of quantum systems (see new reference [51]). Within this framework, the measured quantities — such as heat currents and entropy production rates — retain a consistent thermodynamic interpretation when applied to the targeted microwave modes. Given the central role of the noise-assisted three-level refrigerator as a theoretical model in quantum thermodynamics, we view our experiment as the first – to our knowledge – controlled implementation of a system that captures its essential dynamical features under experimentally tunable conditions.

We hope this clarification demonstrates our intention to present the results with appropriate conceptual framing. While we regret that we did not initially meet the reviewer’s expectations in terms of positioning and terminology, we believe the revised manuscript now accurately reflects the nature and scope of our contribution to the field.

Revisions

We have addressed the referee’s concerns thoroughly throughout the whole manuscript:

1. Removed all references to a ”Brownian refrigerator” in the context of the results.
2. Reframed the results as an analogue quantum simulation of the operating principles of a noise-assisted three-level quantum refrigerator. This framing emphasizes that our measurements reproduce the predicted energy-flow dynamics of the theoretical model and aligns the work with the established concept of analogue quantum simulation. For example, we now explicitly state in the manuscript: *”We frame our results as an analogue quantum simulation of a noise-assisted refrigerator.”*
3. Clarified that the thermodynamic interpretation is applied within an effective model in which only the engineered reservoirs and the system participate in the dynamics. This assumption — that no additional degrees of freedom contribute to the energy exchange — is consistent with the framework of an analogue simulations of a idealized quantum thermal machines.

Reply to Reviewer 2

I have now read the revision from the authors and now satisfied with the latest revision, thus recommend publication.

We thank the reviewer for their careful examination of the manuscript and positive response.

Reply to Reviewer 3

1) The result is seemingly very relevant, but I tend to agree with Referee I in that the presentation is overstating the result and partly misleading. The manuscript purportedly demonstrates “a novel quantum thermal machine that leverages noise-assisted quantum transport to enable a cooling engine in steady state“. But, as far as I understand, no physical element is being refrigerated, so this circuit is not really a cooling engine. Instead, the authors seem to demonstrate energy flows under very specific conditions that would correspond with refrigeration if no further degrees of freedom would exist in the system. This is a very relevant result, but I would rather consider it to fall within the context of quantum analog simulators. The noise-assisted three-level refrigerator is a known toy model in the quantum thermodynamics literature and this is possibly the first experimental implementation approaching the model.

We thank the reviewer for their careful examination of the manuscript and for the insightful suggestion to place the results within the framework of analogue quantum simulations. We acknowledge the concern regarding the original phrasing, which may have overstated the physical interpretation of our device as a refrigerator in the conventional thermodynamic sense.

In the revised manuscript, we have clarified that our work constitutes an analogue quantum simulation of a noise-assisted three-level quantum refrigerator. Specifically, we now frame our results as a demonstration of energy flow dynamics analogous to those of an idealized three-level quantum thermal machine, rather than the operation of a physical cooling engine acting on a macroscopic thermal load. In our experiment, the observed refrigeration corresponds to a measurable reduction in the mean photon occupation of spectrally defined propagating microwave modes — in particular, those coupled to the antisymmetric mode of the artificial molecule. This is directly analogous to well-established cases in quantum thermodynamics and quantum optics, such as optomechanical sideband cooling, where a single mechanical mode (rather than a bulk object) is cooled.

We believe this revised framing aligns the manuscript with experimental quantum thermodynamics and more accurately reflects the conceptual contribution of our work: the first – to our knowledge – experimental demonstration of the key dynamical features of a noise-assisted three-level quantum refrigerator within a superconducting quantum platform.

Revisions

We have addressed the referee’s concerns thoroughly throughout the whole manuscript:

1. We now explicitly present the experiment as an *analogue quantum simulation* of a noise-assisted three-level quantum refrigerator, observing energy-flow dynamics characteristic of an idealized three-level quantum thermal machine. This revised framing is applied consistently throughout the manuscript.
2. We clearly state that no macroscopic thermal load is cooled. Instead, refrigeration corresponds to a measurable reduction in the mean photon occupation of spectrally lo-

calized propagating microwave modes, particularly those coupled to the antisymmetric mode of the artificial molecule.

3. We clarify that the thermodynamic interpretation is valid within an effective model. This is consistent with the standard framework for analogue simulations of idealized three-level quantum thermal machines.

2) It is not clear at all to me how negative powers are calculated, and this is a crucial aspect of the results. The supplementary material states: "The power is directly obtained as the integral of the recorded time trace $f(t)$ squared from each waveguide, in accordance with Parseval's theorem. This is computationally inexpensive and is calibrated against the integral of a known PSD measurement such as the Mollow triplet." How can the sum of absolute values squared ever provide a negative number?

We thank the reviewer for pointing out that the current explanation in the supplementary material is inadequate. To isolate the power emitted by the device, we perform interleaved measurements with and without dephasing applied to the system, such that the emitted power corresponds to the difference $P_{on} - P_{off}$. A positive value of this difference indicates that the device emits photons into the waveguide when activated, whereas a negative value signifies that energy is being absorbed.

Revisions

We have clarified the interleaved nature of the measurement and that the quantity of interest is, in fact, the difference between two measured powers.

3) Also, after equation S1 the following statement reads: "At low power the reflectance goes towards a near unit circle in the IQ plane while reducing towards a single point for lower powers." Do you mean "for higher powers"?

Yes, it should read "for higher powers". Thank you for noticing this.

Revisions

The sentence has been changed state that it is for "for higher powers".

We address the specific comments of the Reviewers point by point below, where we have quoted the Reviewer in **bold font** and replied in normal font.

Reply to Reviewer 3

I have read the revised manuscript, the authors' rebuttal, and the correspondence from the previous review rounds. I understand that my role is to adjudicate the manuscript, particularly in light of the concerns raised by Reviewer 1 and Reviewer 3 regarding the scope of the claims. The point was to reframe the work as an analogue quantum simulation of a noise-assisted refrigerator, rather than the demonstration of a physical cooling device. In my assessment, the authors have mostly resolved the previous concerns. The revision to reframe the results as an analogue quantum simulation positions the work appropriately. By removing claims of demonstrating a "Brownian refrigerator" and clarifying that the observed effect is a reduction in the photon occupation of microwave modes rather than the cooling of a physical object, the manuscript is now clearer and more precise. I believe that the experimental work itself is of high quality, demonstrating control over the system and sensitivity in measuring attowatt-scale heat currents. While I have to admit that I am not really an expert in the field of quantum thermodynamics, the experimental demonstration of the core energy-flow dynamics of a noise-assisted three-level quantum refrigerator model seems to be a timely contribution to the field and also of broad interest, and I believe the manuscript is now well-suited for publication in Nature Communications.

We thank the reviewer for their careful examination of the manuscript and positive response.

1) In the Discussion section, the authors could briefly elaborate on the precise mapping between their experimental system (a superconducting circuit with engineered reservoirs) and the idealized theoretical model it simulates (a three-level system coupled to ideal thermal baths). A clearer statement on which features of the ideal model are captured and which are not would be helpful.

We thank the reviewer for this helpful suggestion. We have slightly revised the Discussion to make the mapping between the experimental implementation and the ideal three-level noise powered refrigerator model explicit. This revision, in addition to the already stated nature of the heat baths, makes a clearer statement on which features of the ideal model are captured and which are not.

Revisions

1. We have added the following to the second paragraph of the discussion: "In our im-

plementation, the three levels correspond to the ground state and the first excited symmetric and antisymmetric states of the artificial molecule. The two microwave waveguides realize the cold and hot baths by selectively coupling to the corresponding transitions. Finally, the engineered dephasing channel acts as an effective infinite-temperature bath by inducing incoherent transitions between the two excited states.”.

2) The manuscript identifies that the dephasing noise acts as a resource, analogous to an infinite-temperature bath, that enables the device to function. For readers less familiar with this specific model, it would be helpful to include a sentence or two clarifying why this incoherent channel can be interpreted as the ”work” input in the context of an absorption refrigerator.

We thank the reviewer for this suggestion. In the standard three-level absorption-refrigerator model, the “work” bath is an infinite-temperature reservoir that drives incoherent transitions on the work transition and thereby supplies the energy quanta needed to power the cooling cycle. In our device, the engineered dephasing noise plays this role within the three-level Lindblad description: it acts as an effective infinite-temperature bath that induces incoherent transitions between the excited symmetric and antisymmetric states, and thus provides the energy that allows heat to flow from the cold to the hot bath. To make this interpretation explicit for readers less familiar with the model, we have added a clarifying sentence to the Discussion.

Revisions

1. Continuing the newly added part of the second paragraph of the discussion: “When the device operates as a refrigerator, the noise provides the energy quanta needed to bridge the energy gap between the symmetric and antisymmetric levels. In this way, it enables population transfer against the thermal gradient and thereby implements the work input in the absorption-refrigerator model.”

3) In the Discussion section, the sentence ending ”...analogue quantum simulation [?] of a noise-assisted refrigerator” appears to have a placeholder for a citation.

Yes, there was indeed a missing citation. Thank you for pointing this out.